# Built Environments And Child Health in WalEs and AuStralia (BEACHES): a study protocol

Rebecca Pedrick-Case [1], Rowena Bailey,[1] Ben Beck [2], Bridget Beesley [3], Bryan Boruff [4], Sinead Brophy [1], Donna Cross [3,5], Gursimran Dhamrait [5,6], John Duncan [4], Peter Gething [5,7], Rhodri D Johnson [1], Ronan A Lyons [1], Amy Mizen [1], Kevin Murray [3], Theodora Pouliou [1], James Rafferty [1], Trina Robinson [5], Michael Rosenberg [8], Jasper Schipperijn [9], Daniel A Thompson [1], Stewart G Trost [10], Alan Watkins [1], Gareth Stratton [11], Richard Fry [1], Hayley Christian [3,5], Lucy J Griffiths [1]

GS, RF, HC and LJG are joint senior authors.

**Correspondence to**
Dr Lucy J Griffiths;
lucy.griffiths@swansea.ac.uk

## ABSTRACT

**Introduction** Childhood obesity and physical inactivity are two of the most significant modifiable risk factors for the prevention of non-communicable diseases (NCDs). Yet, a third of children in Wales and Australia are overweight or obese, and only 20% of UK and Australian children are sufficiently active. The purpose of the Built Environments And Child Health in WalEs and AuStralia (BEACHES) study is to identify and understand how complex and interacting factors in the built environment influence modifiable risk factors for NCDs across childhood.

**Methods and analysis** This is an observational study using data from five established cohorts from Wales and Australia: (1) Wales Electronic Cohort for Children; (2) Millennium Cohort Study; (3) PLAY Spaces and Environments for Children's Physical Activity study; (4) The ORIGINS Project; and (5) Growing Up in Australia: the Longitudinal Study of Australian Children. The study will incorporate a comprehensive suite of longitudinal quantitative data (surveys, anthropometry, accelerometry, and Geographic Information Systems data) to understand how the built environment influences children's modifiable risk factors for NCDs (body mass index, physical activity, sedentary behaviour and diet).

**Ethics and dissemination** This study has received the following approvals: University of Western Australia Human Research Ethics Committee (2020/ET000353), Ramsay Human Research Ethics Committee (under review) and Swansea University Information Governance Review Panel (Project ID: 1001). Findings will be reported to the following: (1) funding bodies, research institutes and hospitals supporting the BEACHES project; (2) parents and children; (3) school management teams; (4) existing and new industry partner networks; (5) federal, state and local governments to inform policy; as well as (6) presented at local, national and international conferences; and (7) disseminated by peer-reviewed publications.

## STRENGTHS AND LIMITATIONS OF THIS STUDY

⇒ The Built Environments And Child Health in WalEs and AuStralia (BEACHES) project uses large representative samples of children from five Wales and Australian cohort studies.
⇒ Standardised built environment measures will be applied across Wales and Australia and linked to the cohort study data at the individual level.
⇒ The contrasting time points, climates, geographies and policy approaches in Wales and Australia will provide stronger evidence of the causal pathways between the built environment and child health.
⇒ The use of existing cohort data sets can limit the use of consistent health outcomes across studies.
⇒ Analyses using routinely recorded data (Wales) may omit some unknown confounders, thereby introducing a moderate level of bias due to confounding.

## INTRODUCTION

Childhood obesity and physical inactivity are two of the most significant modifiable risk factors for non-communicable disease (NCD) prevention in children.[1] The 2017 Commission on Ending Childhood Obesity emphasised that the prevention of modifiable risk factors for NCDs should start as early as possible.[2] Yet only 20% of UK and Australian children are sufficiently active, and over 60% engage in excessive sedentary time, with a third overweight or obese.[1 3]

The built environment in which we live is integral to human health. Research has shown that residing in 'liveable' neighbourhoods characterised by good access to shops, services, quality parks, connected streets to facilitate walking, sufficient residential densities to support public transport services and local businesses, minimal crime and traffic and social connectedness opportunities is associated with improved health outcomes.[4–9] Despite the increasing evidence

of the association between the built environment and physical activity, there is still a paucity of longitudinal research examining the role of the built environment in promoting child health.[10 11] Critical evidence gaps for young people include: (1) causal evidence of the overall impact the built environment has on children's modifiable risk factors for NCDs (ie, physical inactivity, sedentariness and poor diet)[11 12]; (2) the specific effect of individual, familial and combinations of built environment attributes; (3) how these effects vary across different ages of children[13]; (4) how these effects are moderated by socioeconomic status[14]; (5) the mediating role of physical activity and healthy eating on the relationship between the built environment and child obesity[15]; and (6) how the influence of the built environment on children's modifiable risk factors for NCDs varies by different geographical locations.[16] These evidence gaps hinder the formulation of specific, actionable policies to improve the built environment for child health.

The Built Environments and Child Health in WalEs And AuStralia (BEACHES) project is a collaboration between academic institutions in Wales and Australia. This project aims to address identified evidence gaps by using Wales and Australian longitudinal population linked and cohort data to identify and understand how complex and interacting factors in the built environment influence modifiable risk factors for NCDs (physical inactivity, sedentariness, poor diet) across childhood. Five large Welsh and Australian cohort studies, with detailed anthropometric, physical (in)activity, diet and contextual data will be used. We will use the highest quality available spatial data and geospatial techniques to construct an internationally standardised set of metrics, such as walkability, that characterise the built environments each child has inhabited during childhood. A statistical modelling framework will be used to quantify the influence that different built environment characteristics have on children's body mass index (BMI), and the respective roles of physical activity, sedentary behaviour and diet in this relationship.

## Objectives

The purpose of this study is to identify and understand how factors in the built environment influence modifiable risk factors for NCDs across childhood. The objectives are to:

1. Develop a comprehensive Geographic Information Systems (GIS) model of child-specific built environment characteristics using standardised methods for Wales and Australia.
2. Link standardised GIS models of the built environment to e-cohort and standard cohort data for children in Wales and Australia.
3. Determine the direct and indirect (and mediating) relationships between the built environment and: (a) children's modifiable risk factors for NCDs (physical activity, sedentary time and diet), and (b) obesity.

4. Identify how relationships between the built environment and these NCD risk factors vary by children's socioeconomic position and geographic location (across and within Wales and Australia).
5. Produce evidence which policy makers and other stakeholders can use to modify the built environment to enable physical activity and reduce childhood obesity.

## METHODS AND ANALYSIS

This multisite study (funding period: 2020–2024) will use population-level cohort data (including but not limited to surveys, anthropometry, accelerometry, GIS data and biological samples) to address these five objectives (figure 1). We will review the policy landscape and literature on the key built environment and child health-related policies at national and local levels in Wales and Australia. Based on the findings, we will curate and harmonise environmental data that will be used to develop high-resolution spatial models of the built environment. These built environment measures will be linked with cohort and population-level health data (Wales) to identify and understand BMI across childhood using a series of covariate-adjusted, multilevel regression models. Finally, we will communicate to policy makers and planners which modifiable aspects of the built environment may contribute to a reduction in risk factors for NCDs and BMI across childhood.

## Policy and literature review

We will actively engage with stakeholders and carry out a policy landscape review (for the duration of the cohort studies) to examine key built environment and child health-related policies at national and local levels in Wales and Australia. Findings will provide the overall context for policy gaps and landscapes, and the codesign of research to address these gaps, develop best practice guidelines and, overall, reduce NCDs in children.

Our stakeholders include health service commissioners, government departments of planning, transport and health, planning officials, urban planners, building industry, developers and key advocacy non-governmental organisations, as well as third sector and voluntary agencies.

In Wales, we will engage with our stakeholders at three time points throughout the project. These include, for example, Public Health Wales, Active Healthy Kids Wales and Play Wales. We will use appropriate frameworks (eg, 'appreciative inquiry')[17 18] to inform and interpret our quantitative research findings into stakeholder engagement activities. We will map out policies and key legislation across Wales using the six priority areas of the built and natural environments that Public Health Wales has published in the 'Creating healthier places and spaces for our present and future generations' report.[19]

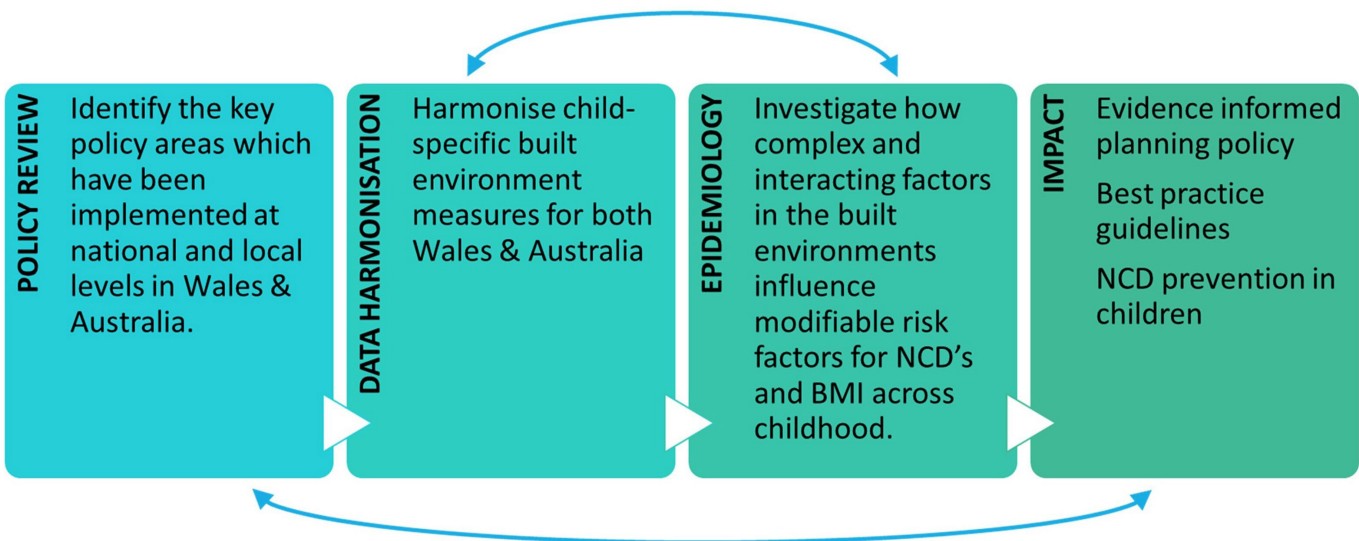

**Partner involvement and input**

**Figure 1** Built Environments and Child Health in Wales and Australia (BEACHES) project workflow. BMI, body mass index; NCD, non-communicable disease.

In Australia, we plan to meet our government and non-government partners quarterly each year for the length of the grant. These include the Western Australian (WA) Department of Local Government, Sport and Cultural Industries, WA Department of Health, WA Department of Transport, WA Local Government Association, Australian Childcare Alliance, Nature Play Australia, Heart Foundation, the PLAY Spaces and Environments for Children's Physical Activity (PLAYCE) partners, Cancer Council WA, Goodstart Early Learning and Hames Sharley. With our partners, we will conduct a policy analysis to investigate how WA and national policies address the health of children through the built environment's influence on obesity and the modifiable risk factors for obesity, physical activity, sedentary behaviour and diet. Policy analysis is crucial to achieving reforms in health promotion by raising awareness of current policy gaps and opportunities and demonstrating successful policy-related actions being taken across the system. The Comprehensive Analysis of Policy on Physical Activity framework will be used to guide the analysis.[20]

**Built environment measures**

GIS-derived built environment measures will be calculated at the residential address level for Wales and Australia. Table 1 outlines the standardised built environment measures that will be created for all Wales and Australian data sets. We will use rich vector spatial data sets which define land utilisation (eg, building footprint and height, cadastre, road centreline and reserve, and points of interest), planning data (eg, active travel routes, open space assessments, fast-food outlets) and earth observation data (eg, Landsat, Sentinel, aerial photography). We have developed a broad range of longitudinal data and methods which characterise the built environment of the two study areas (table 2). These include neighbourhood walkability, garden size/home outdoor area, blue-green space availability,[21] access to services and facilities and fast-food outlets around the home and school.

Annual acquisition and cataloguing of WA GIS data has occurred since 2005, first under the guise of the former University of Western Australia (UWA) Centre for the Built Environment and Health and currently as part of the PLAYCE cohort study. Consistent historic GIS data made available to the BEACHES project have been obtained from Landgate, WA's state geospatial data provider (eg, road centreline, cadastre, street address and imagery), as well as high-resolution four-band aerial imagery from the Urban Monitor[22] aerial imagery acquisition programme (2007, 2009, 2014, 2016, 2018 and 2020), Australian Bureau of Statistics bidecadal census available online since 1996 and historic satellite imagery (table 1).

In Wales, these data will be sourced, where possible, from open data (eg, OpenStreetMap) but will also be supplemented with existing Ordnance Survey data including MasterMap (temporal coverage: 1999 to present day), MasterMap Highways (1997 to present day), MasterMap Imagery Layer (1998 to present day) and AddressBase Premium (1970 to present day), which are made available to the project via the Public Sector GeoSpatial Agreement and Higher Education Agreements with Ordnance Survey. These data, coupled with open-source Landsat and OpenStreetMap data, will allow us to build a rich temporal coverage of the built environment to align with the linked health data.

In previous studies of built environment measures for physical activity, how the neighbourhood is defined has

**Table 1** Wales and Australian standardised measures of the built environment

| Built environment feature | Indicative BE variables | Data source | |
| --- | --- | --- | --- |
| | | **Wales** | **Australia** |
| Outdoor home environment | Home garden/yard space and vegetation | Aerial imagery, NDVI/EVI | Aerial imagery, NDVI/EVI, Urban Monitor, Landgate (cadastre) |
| Child-relevant walkability | Road traffic exposure, dwelling density, residence type, street connectivity, one-way node count, land use mix, 'local living score', public transport stops | OS MasterMap, OSM | Landgate, Main Roads Western Australia, Western Australian Public Transport Authority, Valuer General Office, OSM |
| Child-relevant green/blue space | Public open space type (park, school grounds, natural, club/playgrounds, residual green space), park size category (pocket/small, medium/large, district/regional), park features and attractiveness, vegetation, blue space (distance to coast) | OS MasterMap and green space, OSM | University of Western Australia–Centre for the Built Environment and Health, Landgate |
| Child-relevant destinations and features | Playgroups, childcare centres, before and after school care services, kindergarten/nurseries, primary school, secondary school, other | Local authority data, OSM | Playgroup WA, Western Australian Department of Local Government, Sport and Cultural Industries, Department of Education, Department of Transport, Department of Planning, Lands and Heritage, Department of Health |
| Socioeconomic status | Such as household level of income | WIMD, individual-level SES data in SAIL Databank | Socio-Economic Indexes for Areas (SEIFA) |
| Household location | Required to derive other measures | UPRN (OS), linked to RALF within SAIL database | Property Street Address (PSA), GNAF |

BE, Built Environment; EVI, Enhanced Vegetation Index; GNAF, Geoscape Geocoded National Address File; NDVI, Normalised Difference Vegetation Index; OS, Ordnance Survey; OSM, OpenStreetMap; RALF, Residential Anonymised Linking Field; SAIL, Secure Anonymised Information Linkage; SES, socioeconomic status; UPRN, Unique Property Reference Number; WIMD, Welsh Index of Multiple Deprivation.

**Table 2** Overview of Wales and Australian health data sets

| Characteristics | Data sets and location | | | | |
| --- | --- | --- | --- | --- | --- |
| | WECC (Wales) | MCS (Wales) | PLAYCE (AU) | ORIGINS (AU) | LSAC (AU) |
| Age | 0–17 | 5–14 | 2–9 | 1–3 | 8–17 |
| Sample size | 1.3 million* | 1699–2181 | 2028 | 2036 | 3645 |
| Time points (n) | Dynamic | 4 | 3 | 2 | 5 |
| Collection years | 1990–ongoing† | 2006–2015 | 2015–2022 | 2018–2022 | 2008–2018 |
| BMI | ✓ | ✓ | ✓ | ✓ | ✓ |
| Physical activity | ✕ | ✓ | ✓ | ✓ | ✓ |
| Sedentary time | ✕ | ✓ | ✓ | ✓ | ✓ |
| Diet | ✕ | ✓ | ✓ | ✓ | ✓ |
| Confounders | ✓ | ✓ | ✓ | ✓ | ✓ |
| Geolocation | Home, school | Home, school | Home, childcare, school | Home | Home |

*Cohort size for the analyses will be smaller (as described).
†As this is routine data it is continually collected; however, the observation period for analysis will be selected on availability of data.
AU, Australia; BMI, body mass index; LSAC, Longitudinal Study of Australian Children; MCS, Millennium Cohort Study; PLAYCE, PLAY Spaces and Environments for Children's Physical Activity; WECC, Wales Electronic Cohort for Children.

greatly influenced the results.[23] For example, access to green and blue spaces or destinations (retail and food outlets) can be based on distance from a child's residence measured as a straight line (Euclidean distance) from a household to a destination or based on movement along a road network, with often very different results. Therefore, the GIS and epidemiological teams will work together closely with a 'behaviour-led approach' to define the neighbourhood sizes most likely to correlate with the behaviour of interest by age group; for example, active travel from home to school.

### Harmonisation of built environment measures

As far as reasonably possible, the data, spatial analytical approaches and built environment measures will be standardised and harmonised across the Welsh and Australian study cohorts to enable the calculation of comparable measures. This will involve using temporally aligned GIS and cohort data (ie, built environment metrics generated using spatially referenced data relating to the epidemiological time point of interest), age-specific built environment profiles and neighbourhood characterisations. In addition, feature-level metadata inherent to baseline GIS data sets allows for extraction of historic representations of geographic features to match cohort data collection time points. Where temporal mismatches occur, we will explore methods to impute exposure metrics using the data available, to align with the appropriate time points. Standardisation and harmonisation will increase confidence in the comparability of measures by minimising potential errors from heterogeneity in data collection procedures, software or methodological approaches; and are particularly important in projects between two countries where definitions, symbology and projection in source cartographic data will vary.[24]

While both teams have access to a rich history of local data and built environment measure development, where feasible, we will use open data (eg, OpenStreetMap) and open-source software, such as QGIS, SQL, Python or R, to reduce barriers to reproducing this research in other contexts, thus increasing the potential impact. Methods and code will be made available in future publications, and variations in terminology will be mapped out in a typology with definitions. This will help us identify where broader policy approaches can be taken and where more contextually specific policies will need to be implemented.[25]

### Data sources, linkage and statistical analysis (epidemiology)

Built environment models (see above) will be linked with child cohort data to investigate how multiple factors in the built environments may influence modifiable risk factors for NCDs and BMI across childhood.

### Data sources

Five large Welsh and Australian cohort studies will be used (table 2). They include: (1) Wales Electronic Cohort for Children (WECC); (2) Millennium Cohort Study (MCS Wales); (3) PLAYCE study (Australia); (4) The ORIGINS Project (Australia); and (5) Growing Up in Australia: the Longitudinal Study of Australian Children (LSAC). Further details about each of the data sets are provided below. The age range covers both childhood and adolescence, with multiple data collection dates for each cohort study.

### Wales

#### Wales Electronic Cohort for Children

The Wales Electronic Cohort for Children (WECC) is a total population anonymised electronic cohort study of

all children born and living in Wales since 1990 (table 2) (n=1.3 million). WECC includes information for all live births (approximately 38 000 births per year), Welsh resident mother at birth or a child who becomes a resident in Wales as identified by registering with a general practitioner (GP) in Wales. The primary analyses for this study will be carried out on a smaller cohort selected from WECC who also have a BMI record in the Child Measurement Programme in Wales. This collected measured heights and weights of schoolchildren on entry to primary school (aged 4–5 years) between 2011 and 2019. Approximately 244 000 children were measured during this period. In addition, this cohort will be supplemented with population-level health data (eg, GP records) for children and young people under the age of 18 with heights, weights or BMI values in their health records (up until 2019 approximately 247 000 children have at least one measure of BMI). However, WECC also includes linkage to total population education data and to a subset of child-reported measures (20 000+) including diet, physical activity/physical literacy, well-being and child-assessed rating of local area for safety/play. A fuller description of the WECC and its contents can be found in Hyatt *et al*.[26]

### Millennium Cohort Study

The Millennium Cohort Study (MCS) is a multidisciplinary research project following the lives of around 19 000 young people born across England, Scotland, Wales and Northern Ireland in 2000–2002.[27] The original cohort comprised 18 818 children (72% of those approached) whose parents were first interviewed at home when their child was around 9 months of age. Since then, data have been collected regularly throughout childhood and adolescence. This research aims to use information relating to approximately 2000 children and young people resident in Wales at ages 5 (2006), 7 (2008), 11 (2012) and 14 years (2015). Children's height, weight and waist circumference were measured using standard protocols.[28] Physical activity was measured using accelerometry at age 7[29] and, across all age groups, parent-reported measures of sedentary (eg, parent-reported screen time) and physical activity behaviours (eg, sports club and playing outside),[30 31] as well as dietary factors (eg, snacking, breakfast consumption) and a range of covariate information relating to socioeconomic and demographic factors of the cohort and both resident parents were recorded.

### *Australia*
### The PLAYCE Cohort Study (Western Australia)

The Play Spaces and Environments for Children's Physical Activity (PLAYCE), Western Australia, is an ongoing cohort study (2015 to current), investigating the influence of the childcare, home and neighbourhoodbuilt environments (parent perceptions and objectively measured) on children's physical activity, sedentary behaviour, diet and weight status.[32] Baseline data were collected for 2028 children, aged 2–5 years and recruited from 118 childcare centres (response rate 46%) in metropolitan Perth, WA. Childcare centres were recruited evenly across socioeconomic status and size (small and large). Two additional data time points were collected once children commenced formal schooling at 5–7 years (2019–2020) and 7–9 years (2020–2022). The study design enables modifiable risk factors to be examined as children transition between different built environment settings: home, childcare; and full-time school. Full details of the PLAYCE study protocol have been published.[33]

Children's height, weight and waist circumference were measured by the research team using standard protocols. Physical activity was measured using accelerometry, and parent-reported structured and unstructured physical activities including outdoor play.[33–36] Sedentary behaviour was measured using accelerometry (time spent sedentary) and parent-reported screen time.[33–35] Diet was assessed using a parent-reported short Food Frequency Questionnaire.[37]

### The ORIGINS Project (Western Australia)

ORIGINS is a collaborative initiative between the Telethon Kids Institute and Joondalup Health Campus. ORIGINS aims to recruit a birth cohort of 10 000 families enrolled during pregnancy and followed children from birth to age 5.[38] ORIGINS is fully integrated into clinical and diagnostic services. It aims to improve understanding of the early antecedent pathways that mediate NCD development through the study of early environments, maternal and paternal physical health and genetics. ORIGINS' overarching goal is to reduce the rising epidemic of NCDs by providing children with 'a healthy start to life'. Data collected from active participants aged 1 year (2018–2020) and 3 years (2020–2022) will be included. Children's height, weight and waist circumference were measured by trained staff. Physical activity was measured using parent-reported structured and unstructured physical activities as well as outdoor play.[36] Sedentary behaviour was measured via parent-reported screen time. Parents reported their child's diet using the Australian Eating Survey; a validated semiquantitative Food Frequency Questionnaire.[39]

### LSAC (Australia)

Growing Up in Australia: the LSAC (Longitudinal Study of Australian Children) is an ongoing prospective study that used a two-cohort cross-sequential design and in 2004 recruited 5107 children to the baby cohort (~9 months) and 4983 children to the kindergarten cohort (~4.8 years).[40] The sampling method provided a representative sample of Australian children. Data were collected every 2 years for both cohorts. Data from children aged 8–17 years (8–9; 10–11; 12–13; 14–15; and 16–17) from Perth, Sydney and Melbourne metropolitan areas, collected between 2008 and 2018, will be included (n=3645). Full details of the LSAC study are published elsewhere.[40]

Children's height, weight, body composition and waist circumference were measured by parents. Physical activity was measured using parent-reported time use diaries which included structured and unstructured physical activities. Sedentary behaviour was gathered through parent reports and the children's time use diary for three types of screen-based activities (television viewing, computer use and electronic games).[41] Each parent reported their child's diet via food diaries. In the Child Health CheckPoint, children completed a questionnaire regarding their usual intake of various foods including fruit and vegetable consumption, water intake and breakfast consumption.

## Data linkage

### Wales

Routinely collected health data and cohort study data from studies, such as the MCS, are available via the Secure Anonymised Information Linkage (SAIL) Databank.[42 43] This is a unique resource of linked longitudinal health, socioeconomic and environmental data relating to the people of Wales. One of the core linkage mechanisms is the Residential Anonymised Linking Field (RALF).[44] The benefits of the RALF system are threefold. First, it allows the GIS modelling of population-level socioenvironmental measures for approximately 1.4 million households in Wales at ±1 m spatial resolution. Second, it allows the grouping of individuals into household units in SAIL, thus reducing ecological fallacies and the impacts of the modifiable areal unit problem. Third, it allows house moves to be identified and hence length of exposure to the built environment to be examined.

### Australia

Home addresses will be geocoded for all three Australian cohorts (ORIGINS, PLAYCE and LSAC) to enable the creation of GIS measures of the natural and built environments by the Australian team.[33 45] PLAYCE and ORIGINS cohort data will be linked to built environment measures using a unique participant identification number. Where possible, change in addresses over time will be examined for these cohorts. LSAC data linkage will involve the Australian Bureau of Statistics, the Australian Institute of Family Studies Data Linkage and Integrating Authority, the Australian Institute of Family Studies Data Management team and the National Centre for Longitudinal Data.

## Statistical analysis

A series of covariate-adjusted multilevel regression models will be fitted to explore the impact of different aspects of the built environment on BMI/NCD risk factors, to quantify the effect of each associative relationship using a common outcome variable. A multilevel model structure will be used to adjust for inherent autocorrelation that exists within groups of individuals. Covariates will include, for example, child age, sex, socioeconomic status (eg, parent education, neighbourhood-level disadvantage)

and family composition. We will also examine how these relationships vary by different population groups (eg, young children, school-age children and adolescents). This series of analyses will enable a comparison and prioritisation of the built environment factors most strongly associated with BMI and quantify the magnitude of variation in effect size across various built environment features. We will then standardise these results to rank characteristics of the built environment which may contribute to better health, to assist prioritisation of public services and resources.

The direct, indirect and total effects of the built environment on children's BMI and other NCD risk factors will be estimated using well-established multilevel mediation modelling. We will also determine if differences in these factors between advantaged and disadvantaged neighbourhoods can be explained in part by differences in built environment attributes (eg, better or poorer access to green space). The mediation effect of physical activity and the moderation effect of neighbourhood socioeconomic disadvantage on the relationship between the built environment and BMI will initially be tested as separate models. A moderated path analysis using all longitudinal data will combine moderation and mediation, and identify direct, indirect and total mediation effects, and show how these effects vary across the levels of the moderator. All analyses will adjust for baseline and, where appropriate, temporal confounding. Special attention will also be paid to potential confounders of the outcome (eg, BMI) and the mediating variable (eg, physical activity) being examined. The role of age, sex, socioeconomic status and urban/rural locality will be examined.

Statistical analyses will be conducted separately for Welsh and Australian data sets. A set of built environment measures will be standardised and linked to the five cohort data sets. Data governance prohibits pooling of data sets; however, findings from both countries' analyses of their respective linked health cohort and built environment data sets will be compared to determine the differences between different age groups of children in Wales and Australia.

Formal statistical analysis plans will be drafted in advance of the analyses and agreed with our Expert Advisory Group. These will specify our modelling strategies, rules on dealing with missing data and reporting conventions, including the provision of Consolidated Standards of Reporting Trials diagrams.

## Patient and public involvement

Development of the research questions and outcome measures was informed by ongoing engagement with stakeholders who have an interest in child health, such as Public Health Wales, and the PLAYCE Community Reference Group (WA). This study uses anonymised routinely collected and cohort data, so it was not possible to directly consult the individuals included in the study.

Results will be disseminated by engaging children directly such as via a short video that can be shown in

schools and online, science outreach workshops and existing child health networks to translate the findings into useful advice for children, their families and policy makers. Please refer to the 'Ethics and dissemination' section for further information.

## ETHICS AND DISSEMINATION

All data will be anonymised and, in Wales, linked within the privacy protecting SAIL Databank. We will be using anonymised data and therefore we are exempt from the National Research Ethics Committee. An Information Governance Review Panel (IGRP) (Project ID: 1001) provided approval to link these data. The PLAYCE study has existing Human Research Ethics Committee approval from UWA (2020/ET000353). Ethics approvals have been received from Ramsay Human Research Ethics Committee WA for the ORIGINS and built environment analyses (HREC 2155W) and the UWA Human Research Ethics Committee for the LSAC and built environment secondary analyses (2022/ET000573). All Australian research will abide by the National Health and Medical Research Council's National Statement on Ethical Conduct in Human Research (2008) and is considered a negligible risk.

Our dissemination and impact generation strategies will be informed by research on knowledge mobilisation models,[46] to impact at a variety of scales (including national and local governments and across multiple sectors), and will not be confined to a Welsh-Australian context. Throughout, we will involve stakeholders and policy makers in the research process from inception to translation, seeking their feedback on our process, progress and findings. We will disseminate findings through community engagement (eg, at public events for families or to school management teams) and through existing and new industry partner networks. An information sheet will be prepared in plain English with infographics, and a video explaining the project and its findings will be made available online and advertised through social media. Findings from the study will be published in a comprehensive report and an infographic summary. We will notify all stakeholders and promote publication through international networks established during this research. We will hold seminars and workshops to report our findings to stakeholders and the public.

## DISCUSSION

The purpose of this study is to identify and understand how complex and interacting factors in the built environment influence modifiable risk factors for NCDs across childhood. Harmonising both child and built environment indicators across five large-scale studies enables analysis of an effectively larger sample size, spans a broader set of age ranges and leverages greater heterogeneity in built environments. By harmonising analyses from Welsh and Australian settings, the BEACHES project provides

a unique opportunity to identify impacts of the built environment that are common across settings and where place-specific, physical, cultural and policy environments mediate these effects.

We aim to communicate to policy and planners which modifiable aspects of the built environment result in the most significant reductions in the risk factors for NCDs across childhood. This will enable action at both a national and international scale and, in turn, will contribute to population-level reductions in NCDs. This is an essential component of our work to be able to bring about large-scale improvement in the built environment to reduce NCDs beginning in early childhood. Furthermore, the outputs of this research could be used to inform future research by aggregating evidence of individual associations in a single dynamic model.

**Author affiliations**
[1]Population Data Science, Swansea University Medical School, Swansea, UK
[2]School of Public Health and Preventive Medicine, Monash University, Melbourne, Victoria, Australia
[3]School of Population and Global Health, University of Western Australia, Perth, Western Australia, Australia
[4]School of Agriculture and Environment, University of Western Australia, Perth, Western Australia, Australia
[5]Telethon Kids Institute, University of Western Australia, Perth, Western Australia, Australia
[6]School of Health and Society, University of Wollongong, Wollongong, New South Wales, Australia
[7]Faculty of Health Sciences, Curtin University, Perth, Western Australia, Australia
[8]School of Human Sciences, University of Western Australia, Perth, Western Australia, Australia
[9]Department of Sports Science and Clinical Biomechanics, University of Southern Denmark, Odense, Denmark
[10]School of Human Movement and Nutrition Sciences, The University of Queensland, Brisbane, Queensland, Australia
[11]Research Centre in Applied Sports, Technology, Exercise and Medicine, Swansea University, Swansea, UK

**Acknowledgements** This is a joint initiative between the Telethon Kids Institute, The University of Western Australia, and Swansea University, with collaborators from Curtin University, Monash University, The Queensland University of Technology, University of Southern Denmark. Wales: We would like to acknowledge all the data providers who make anonymised data available for research. This study makes use of anonymised data held in the Secure Anonymised Information Linkage (SAIL) Databank, hosted by Swansea University, UK. We would like to thank the Welsh Assembly Government for supporting SAIL and the Centre for Population Health National Centre for Population Health and Wellbeing Research for the work on the Wales Electronic Cohort for Children (WECC). This work uses data provided by patients and collected by the NHS as part of their care and support. We wish to acknowledge the collaborative partnership that enabled acquisition and access to the de-identified data, which led to this output. All research conducted has been completed under the permission and approval of the SAIL independent Information Governance Review Panel (IGRP) project number 1001. We also thank all the Millennium Cohort families for their participation, the director of the Millennium Cohort Study and colleagues in the management team at the Centre for Longitudinal Studies, UCL Institute of Education. Australia: Partners for the Australian arm include the Western Australian (WA) Department of Local Government, Sport and Cultural Industries, WA Department of Health, WA Department of Transport, WA Local Government Association, Australian Childcare Alliance, Nature Play Australia, Heart Foundation, Cancer Council WA, Goodstart Early Learning and Hames Sharley. BEACHES is a subproject of The ORIGINS Project. The ORIGINS Project is only possible because of the commitment of the families in ORIGINS. We are grateful to all the participants, health professionals and researchers who

support the project. We would also like to acknowledge and thank the following teams and individuals who have made The ORIGINS Project possible: The ORIGINS Project team; CEO Dr Kempton Cowan, executive staff and obstetric, neonatal and paediatric teams, Joondalup Health Campus (JHC); Director Professor Jonathan Carapetis and executive staff, Telethon Kids Institute; Mayor Tracey Roberts, City of Wanneroo; Mayor Albert Jacobs, City of Joondalup; Professor Fiona Stanley, patron of ORIGINS; members of ORIGINS Community Reference and Participant Reference Groups; Research Interest Groups and the ORIGINS Scientific Committee. Growing Up in Australia: The Longitudinal Study of Australian Children (LSAC) is conducted in partnership between the Department of Social Services (DSS), the Australian Institute of Family Studies (AIFS) and the Australian Bureau of Statistics (ABS), with advice provided by a consortium of leading researchers from research institutions and universities throughout Australia. This study does not necessarily reflect their views. We thank the children and families who are participating in the LSAC.

**Contributors** HC, GS, LJG and RF are the principal investigators who conceptualised the overall study, established research collaborations and led the project implementation. Coinvestigators who contributed to obtaining the research funding include BBec, BBo, SB, DC, PG, RAL, AM, KM, MR, JS, ST and AW. In addition to the principal investigators and coinvestigators, other team members (RP-C, RB, BBee, JD, RDJ, JR, DAT, TR) are responsible for the development of Geographic Information Systems measures and associated data analysis. RB, KM, AW, RDJ, JR, HC and LJG are responsible for the design and implementation of the statistical analyses. GS, TP, HC and GD are responsible for the policy reviews and stakeholder engagement. RP-C, GD, RF, HC and LJG drafted this manuscript. All authors revised and approved the final version of the manuscript.

**Funding** Wales: The Welsh arm of the BEACHES study was funded by the UK Medical Research Council (MRC; grant number: MR/T039329/1). The Welsh Assembly government fund SAIL and the National Centre for Population Health & Wellbeing Research for their work on the Wales Electronic Cohort for Children (WECC), which provided the data for the work. This work will also be supported by the Administrative Data Research (ADR) Wales, part of the ADR UK investment, which unites research expertise from Swansea University Medical School and the Wales Institute of Social and Economic Research and Data (WISERD) at Cardiff University with analysts from the Welsh Government. ADR UK is funded by the Economic and Social Research Council (ESRC), part of UK Research and Innovation. The Millennium Cohort Study is funded by grants to the Centre for Longitudinal Studies at the Institute of Education from the Economic and Social Research Council and a consortium of government department. Australia: The Australian arm of the BEACHES study was funded by the Australian National Health and Medical Research Council–UKRI-NHMRC Built Environment Prevention Research Scheme (GNT1192764). This research is supported (partially) by the Australian Government through the Australian Research Council's Centre of Excellence for Children and Families over the Life Course (Project ID: CE200100025). HC is supported by an Australian National Heart Foundation Future Leader Fellowship (102549). The ORIGINS Project has received core funding support from the Telethon Perth Children's Hospital Research Fund, Joondalup Health Campus, the Paul Ramsay Foundation and the Commonwealth Government of Australia through the Channel 7 Telethon Trust. Substantial in-kind support has been provided by Telethon Kids Institute and Joondalup Health Campus.

**Competing interests** None declared.

**Patient and public involvement** Patients and/or the public were not involved in the design, or conduct, or reporting, or dissemination plans of this research.

**Patient consent for publication** Not applicable.

**Provenance and peer review** Not commissioned; externally peer reviewed.

**ORCID iDs**
Rebecca Pedrick-Case http://orcid.org/0000-0003-2307-1808
Ben Beck http://orcid.org/0000-0003-3262-5956
Bridget Beesley http://orcid.org/0000-0003-0432-3766
Bryan Boruff http://orcid.org/0000-0001-6693-0671
Sinead Brophy http://orcid.org/0000-0001-7417-2858
Donna Cross http://orcid.org/0000-0001-6217-5058
Gursimran Dhamrait http://orcid.org/0000-0002-5191-211X
John Duncan http://orcid.org/0000-0001-9752-1002
Peter Gething http://orcid.org/0000-0001-6759-5449
Rhodri D Johnson http://orcid.org/0000-0001-9636-0753
Ronan A Lyons http://orcid.org/0000-0001-5225-000X
Amy Mizen http://orcid.org/0000-0001-7516-6767
Kevin Murray http://orcid.org/0000-0002-8856-6046
Theodora Pouliou http://orcid.org/0000-0002-1162-1174
James Rafferty http://orcid.org/0000-0002-1667-7265
Trina Robinson http://orcid.org/0000-0003-2638-958X
Michael Rosenberg http://orcid.org/0000-0002-9950-5775
Jasper Schipperijn http://orcid.org/0000-0002-6558-7610
Daniel A Thompson http://orcid.org/0000-0002-1769-2870
Stewart G Trost http://orcid.org/0000-0001-9587-3944
Alan Watkins http://orcid.org/0000-0003-3804-1943
Gareth Stratton http://orcid.org/0000-0001-5618-0803
Richard Fry http://orcid.org/0000-0002-7968-6679
Hayley Christian http://orcid.org/0000-0001-8486-5746
Lucy J Griffiths http://orcid.org/0000-0001-9230-624X

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
