## [Reviewer comments · BMJ Open]

ARTICLE DETAILS

TITLE (PROVISIONAL)	Built Environment and Child Health in Wales and Australia (BEACHES): a study protocol
AUTHORS	Pedrick-Case, Rebecca; Bailey, Rowena; Beck, Ben; Beesley, Bridget; Boruff, Bryan; Brophy, Sinead; Cross, Donna; Dhamrait, Gursimran; Duncan, John; Gething, Peter; Johnson, Rhodri; Lyons, Ronan; Mizen, Amy; Murray, Kevin; Poulidou, Theodora; Rafferty, James; Robinson, Trina; Rosenberg, Michael; Schipperijn, Jasper; Thompson, Daniel; Trost, Stewart; Watkins, Alan; Stratton, Gareth; Fry, Richard; Christian, Hayley; Griffiths, Lucy

VERSION 1 – REVIEW

REVIEWER	Fuller, Daniel Memorial University
REVIEW RETURNED	14-Apr-2022

GENERAL COMMENTS	Summary: This protocol paper outlines the study design, data collection, and analysis for the BEACHES study, an international study of the longitudinal association between childhood physical activity/BMI as outcomes and built environment characteristics as predictors. The paper uses data from existing cohorts in Wales and Australia. General comment: This is interesting work. I have made specific comments below. One general comment for the authors is that the description of each cohort could be made shorter and more clear if the authors followed a standardized way to present each cohort. At the moment, the descriptions of each cohort vary in detail, descriptions of the samples, and outcome variables. Introduction Comment 1: Page 4 Lines 108-110: I would recommend that the authors write a few sentences about what the evidence does show in this area. Lines 104-108 focus on adults but that space could be used to provide a fair assessment of what the literature shows for children/youth. There is considerable research in this area and at the moment the introduction does not provide sufficient overview of that literature. Methods
---

	Comment 2: Stakeholder engagement. The paper states that in Wales stakeholders will be engaged with at three time points. It is not clear how many engagements will happen in Australia. Can the authors provide for detail about the number and timing of these engagements? Comment 3: Longitudinal study data collection timelines. Related to comment 2, it is unclear what the timelines are for data collection in each of the different cohort studies included in this protocol. Adding a figure with a timeline for each year and which cohort includes data is important for the reader to assess the comparability of these cohorts. Comment 4: Table 2: There might be a rendering problem with the table but I see a bunch of question marks in the table when there should be check marks. Can the authors please check? Comment 5: Built environment data harmonization. It is unclear how the authors will collect/have collected historical built environment data. The majority of the BE data collection rests on Open Street Maps (which is a strength), but some of the cohorts started at 1990, 2006, or 2008. How will the authors collect and harmonize historical BE data? This may have been done but needs to be more clearly reported in the protocol. Comment 6: Harmonization of the outcome variables. Table 2 reports that outcome variables are reported for each study. However, it is not clear if the same or similar measures were used to collect the data. It appears that physical activity was measured using self-report in some instances and accelerometry in others. There is little mention of harmonization of these outcomes. This is critically important step that requires much more detail in the paper. Comment 7: Related to comment 6, it appears that in some cases BMI was only measured once during the cohort study. Perhaps in addition to the timeline I have suggested in comment 3, there could be a table or timeline showing which variables are available in which years, from which cohort. Comment 8: Page 13 Line 30. The authors state that they will use multilevel and machine learning models. Machine learning models are not discussed further in the analysis section. I suggest they authors provide specific detail about the ML models they will use or remove this from the first sentence of the analysis section. Comment 9: There is not mention in the analysis section of how the authors propose to deal with age, period, cohort effects in their analysis. Some discussion of the approach the authors will use is important to for the reader. Comment 10: Page 14: Lines 15-24: I found it very surprising that the authors state that analyses will be conducted separately for Wales and Australia. There is a lot of detailed and important work going into data harmonization in this research. Could the authors provide more and specific detail as to why the data cannot be combined across countries? I found the statement “small differences in the methodology used in each cohort that makes pooling datasets unfeasible” to be unhelpful.
--	--

REVIEWER	Liu, Tao Guangdong Provincial Institute of Public Health
REVIEW RETURNED	24-May-2022

GENERAL COMMENTS	Thanks very much for giving me an opportunity to review this study protocol which aimed to identify and understand how complex and interacting factors in the built environment influence modifiable risk factors for NCDs across childhood using data from five established cohorts from Wales and Australia. This is a well written protocol. However, I have several minor suggestions.  1. In Lines 79-82, the findings may also be reported to school managers and teachers. 2. Harmonisation of built environment measures is crucially important for studies across regions. Therefore, I suggest the authors to give more information of the harmonization approaches. 3. Did these cohort collect children's address changes during the follow-up?
---

VERSION 1 – AUTHOR RESPONSE

Reviewer: 1

This is interesting work. I have made specific comments below. One general comment for the authors is that the description of each cohort could be made shorter and clearer if the authors followed a standardized way to present each cohort. At the moment, the descriptions of each cohort vary in detail, descriptions of the samples, and outcome variables.

Response: Thank you for this recommendation, we have modified these sections as suggested to try and standardise the information provided.

Introduction

Comment 1: Page 4 Lines 108-110: I would recommend that the authors write a few sentences about what the evidence does show in this area. Lines 104-108 focus on adults but that space could be used to provide a fair assessment of what the literature shows for children/youth. There is considerable research in this area and at the moment the introduction does not provide sufficient overview of that literature.

Response: The literature described in predominantly child specific. We have tried to make this clearer throughout the Introduction and have added an additional reference for a review that focuses on children and adults.

Methods

Comment 2: Stakeholder engagement. The paper states that in Wales stakeholders will be engaged with at three time points. It is not clear how many engagements will happen in Australia. Can the authors provide for detail about the number and timing of these engagements?

Response: We have inserted the following information: "In Australia, we plan to meet our government and non-government partners quarterly each year for the length of the grant. These include..."

Comment 3: Longitudinal study data collection timelines. Related to comment 2, it is unclear what the timelines are for data collection in each of the different cohort studies included in this protocol. Adding a figure with a timeline for each year and which cohort includes data is important for the reader to assess the comparability of these cohorts.

Response: Table 2 (page 10) includes the timeline for the data collection period and timepoints for each cohort study, as well as information on the age range, sample size, key outcomes, confounders and geo-location.

Comment 4: Table 2: There might be a rendering problem with the table but I see a bunch of question marks in the table when there should be check marks. Can the authors please check?

Response: This was a formatting issue when our Word document was converted to a pdf during submission. We will check this in our resubmission.

Comment 5: Built environment data harmonization. It is unclear how the authors will collect/have collected historical built environment data. The majority of the BE data collection rests on Open Street Maps (which is a strength), but some of the cohorts started at 1990, 2006, or 2008. How will the authors collect and harmonize historical BE data? This may have been done but needs to be more clearly reported in the protocol.

Response: We agree this is an important aspect of the project and have added more details to the section on built environment harmonisation (page 8)

Comment 6: Harmonization of the outcome variables. Table 2 reports that outcome variables are reported for each study. However, it is not clear if the same or similar measures were used to collect the data. It appears that physical activity was measured using self-report in some instances and accelerometry in others. There is little mention of harmonization of these outcomes. This is critically important step that requires much more detail in the paper.

Response: As now stated on page 8, the built environment measures for all five cohort studies across Australia and Wales will be standardised. The key outcomes variables (e.g., BMI) are based on standardised measures in each cohort study. Each cohort study represents a different age stage of childhood and together will provide a comprehensive picture of the influence of different aspects of the built environment on child modifiable risk factors for NCDs (physical inactivity, sedentary time, dietary intake, BMI). As the key outcome variables have been collected retrospectively and at different points in time, it is not possible to truly harmonise all variables and thus pool datasets for a single combined analysis. No change made.

Comment 7: Related to comment 6, it appears that in some cases BMI was only measured once during the cohort study. Perhaps in addition to the timeline I have suggested in comment 3, there could be a table or timeline showing which variables are available in which years, from which cohort.

Response: Thank you – this information is included in Table 2 and the related text description. BMI is measured at each time point for each cohort study except one – the Wales Electronic Cohort for Children study. This is indicated in the table footnote. We have clarified BMI data availability in WECC: ‘...This will be supplemented with BMI data from the Child Measurement Programme in Wales (measured height and weights at age 5 years) and from population-level health data (general practitioner records, hospital admissions) for those children and young people who have follow-up (post age 5) heights, weights or BMI values in their health records’

Comment 8: Page 13 Line 30. The authors state that they will use multilevel and machine learning models. Machine learning models are not discussed further in the analysis section. I suggest they authors provide specific detail about the ML models they will use or remove this from the first sentence of the analysis section.

Response: Information related to machine learning models has been removed from the text. Thank you for this suggestion.

Comment 9: There is no mention in the analysis section of how the authors propose to deal with age, period, cohort effects in their analysis. Some discussion of the approach the authors will use is important to for the reader.

Response: As outlined on page 14, (also see response to comment 6) analyses will be conducted separately for each cohort study. Thus, it is not necessary to account for cohort effects. Furthermore, lines 369-372 outline that the models will adjust for temporal effects and socio-demographic and other relevant covariates. We also highlight that we will examine how relationships vary by different age groups of children. We have not provided the specific details of all models to be fitted because this is dependent on the cohort study and type of data. A full statistical analysis plan will be made available once the data has been fully captured and harmonised.

Comment 10: Page 14: Lines 15-24: I found it very surprising that the authors state that analyses will be conducted separately for Wales and Australia. There is a lot of detailed and important work going into data harmonization in this research. Could the authors provide more and specific detail as to why the data cannot be combined across countries? I found the statement “small differences in the methodology used in each cohort that makes pooling datasets unfeasible” to be unhelpful.

Response: Please see responses to Comments 6 and 9. Data governance prohibits merger of datasets; we have added this detail to the paper (line 376). We agree that this statement was unhelpful, and so have deleted it. Thank you.

Reviewer: 2

Comment 1: Thanks very much for giving me an opportunity to review this study protocol which aimed to identify and understand how complex and interacting factors in the built environment influence modifiable risk factors for NCDs across childhood using data from five established cohorts from Wales and Australia. This is a well written protocol. However, I have several minor suggestions. In Lines 79-82, the findings may also be reported to school managers and teachers.

Thank you, we agree. This has been added to the text (see page 3, line 80 and page 15, line 413).

Comment 2: Harmonisation of built environment measures is crucially important for studies across regions. Therefore, I suggest the authors to give more information of the harmonization approaches.

Response: Please see response to Reviewer 1 Comments 5 and 10.

Comment 3: Did these cohort collect children’s address changes during the follow-up?

Response: Yes, change in child address at each time point has been collected in some of the cohort studies such as PLAYCE and ORIGINS. The research team are currently investigating the data to determine if there is sufficient sample size to conduct analyses within and between participants who change and do not change their address over time. We have added two lines to page 13 (line 338) to indicate that we will examine house moves where possible (i.e. numbers permitting).

VERSION 2 – REVIEW

REVIEWER	Fuller, Daniel Memorial University
REVIEW RETURNED	25-Jul-2022

GENERAL COMMENTS	Summary: This protocol paper outlines the study design, data collection, and analysis for the BEACHES study, an international study of the longitudinal association between childhood physical activity/BMI as outcomes and built environment characteristics as predictors. The paper uses data from existing cohorts in Wales and Australia. General comment: I thank the authors for the responses to the reviews overall. However, there are a two critical revisions that must be addressed. Comment 1: I do not feel the longitudinal nature of the data is correctly presented in Table 2. I had previously suggested a figure but regardless of how the authors chose to address this more detail is needed. I believe this could immensely improve the presentation of the data and the paper. For example, the WECC cohort in the description says that it is updated every 3 months (Line 282), but the description says collection is ongoing. Will authors take a specific time point of the 4 data collection that happen during the year? The reader is left to do a lot of work. From what I understand: WECC – 1990 – ongoing with participants measured every 3 months... MCS - 2006-2015 with participants measured at ages 5 (2006), 7 (2008), 11 (2012) and 14 years (2015) PLAYCE - 2015-2022 with participants aged 2-5 years measured in 2015, 5-7 years (2019-20) and 7-9 years (2020-21). ORIGINS - 2018-2022 with participants measured in 2018 and then in the text on line 352-353 it reads “In 2021, the cohort will include data on 2,036 children collected at 12 and 36 months”. Table 2 says 2 time points but the text suggests 3 time points? What years are the 12 and 36 month data collection? LSAC - 2008-2018 with 8 time points according to Table 2 but text on lines 363-364 states “Data were collected every two years for both cohorts.” What years were data collected? The authors must improve the presentation. Specifically, it should be easy for the reader to identify the years in which data were collected for each cohort, and which measures were collected in a given year for a given cohort. Comment 2: Related to the harmonization of built environment data. The authors have not addressed the historical question of how these data will be/have been collected. For example, OSM was founded in 2004 and historical data are difficult to obtain/not highly detailed in my experience. Even historical census or other types of built environment data may be difficult to obtain. I feel the authors are oversimplifying the real challenges presented by historical data. I would like to see detail about the how authors will address these challenges and a frank discussion of these challenges.
--

VERSION 2 – AUTHOR RESPONSE

Reviewer: 1

I thank the authors for the responses to the reviews overall. However, there are two critical revisions that must be addressed.

Response: Thank you for acknowledging our previous changes. As set out below we have now revised the manuscript to include the additional information you've suggested.

Comment 1: I do not feel the longitudinal nature of the data is correctly presented in Table 2. I had previously suggested a figure but regardless of how the authors chose to address this more detail is needed. I believe this could immensely improve the presentation of the data and the paper. For example, the WECC cohort in the description says that it is updated every 3 months (Line 282), but the description says collection is ongoing. Will authors take a specific time point of the 4 data collection that happen during the year? The reader is left to do a lot of work. From what I understand: WECC – 1990 – ongoing with participants measured every 3 months... MCS - 2006-2015 with participants measured at ages 5 (2006), 7 (2008), 11 (2012) and 14 years (2015) PLAYCE - 2015-2022 with participants aged 2-5 years measured in 2015, 5-7 years (2019-20) and 7-9 years (2020-21). ORIGINS - 2018-2022 with participants measured in 2018 and then in the text on line 352-353 it reads "In 2021, the cohort will include data on 2,036 children collected at 12 and 36 months". Table 2 says 2 time points but the text suggests 3 time points? What years are the 12- and 36-month data collection? LSAC - 2008-2018 with 8 time points according to Table 2 but text on lines 363-364 states "Data were collected every two years for both cohorts." What years were data collected? The authors must improve the presentation. Specifically, it should be easy for the reader to identify the years in which data were collected for each cohort, and which measures were collected in a given year, for a given cohort.

Response: We are sorry for this lack of clarity. We have amended table two and the related text. The following changes have been made to each dataset description:

WECC data - We have amended the table footnote to make it clearer that we are using routinely collected data, and our analyses will be carried out on a smaller sample who have available measures in the Child Measurement Data and/or health data. We have also revised the text to: "...The primary analyses for this study will be carried out on a smaller Cohort selected from WECC who also have a BMI record in the Child Measurement Programme in Wales. This collected measured height and weights of school children upon entry to primary school (aged 4-5 years) between 2011 and 2019. Approximately 244,000 children were measured during this period. In addition, this Cohort will be supplemented with population-level health data (e.g., general practitioner (GP) records) for children and young people under the age of 18 with heights, weights or BMI values in their health records (up until 2019 approximately 247,000 children have at least one measure of BMI). However, WECC also includes linkage..."

MCS data - No changes made. Information is provided on age and years of data collection.

ORIGINS data - Table two is correct – there are two time points of ORIGINS data. As this is a cohort study there is rolling recruitment and data collection. This has been clarified in the text: 'Data collected from active participants aged one-year old (2018-2020) and three-years old (2020-2022) be included (n=2,036).'

LSAC data - Table two has been updated to indicate there are five waves (timepoints) of LSAC data we will link built environment measures to. The years the data were collected, and ages of children has been clarified in the text: ‘Data were collected every two years for both cohorts. Data from children aged 8-17 years (8-9; 10-11; 12-13; 14-15; and 16-17 years old) from the Perth, Sydney and Melbourne metropolitan areas, collected between 2008 and 2018, will be included (n=3,645). Full details of the LSAC study are published elsewhere.[39]’

Comment 2: Related to the harmonization of built environment data. The authors have not addressed the historical question of how these data will be/have been collected. For example, OSM was founded in 2004 and historical data are difficult to obtain/not highly detailed in my experience. Even historical census or other types of built environment data may be difficult to obtain. I feel the authors are oversimplifying the real challenges presented by historical data. I would like to see detail about the how authors will address these challenges and a frank discussion of these challenges.

Response: We apologise for the lack of clarity around the geospatial data. We have made edits to the manuscript to explicitly state the temporal coverage of the data we will use - beyond OSM. These data are held by the project teams in Wales and Australia or are accessible via existing data access agreements giving us access to timestamped databases. Whilst we therefore agree that there are challenges in different parts of the world in relation to access to rich historical geospatial data, we are fortunate that these data are available in both Wales and Australia and can be used in this project.

VERSION 3 – REVIEW

REVIEWER	Fuller, Daniel Memorial University
REVIEW RETURNED	01-Sep-2022
GENERAL COMMENTS	I thank the authors for the attention to detail regarding the geospatial data section.